# Optical Genome Mapping as a Potential Routine Clinical Diagnostic Method

**DOI:** 10.3390/genes15030342

**Published:** 2024-03-07

**Authors:** Hayk Barseghyan, Doris Eisenreich, Evgenia Lindt, Martin Wendlandt, Florentine Scharf, Anna Benet-Pages, Kai Sendelbach, Teresa Neuhann, Angela Abicht, Elke Holinski-Feder, Udo Koehler

**Affiliations:** 1Medical Genetics Center (MGZ), 80335 Munich, Germany; 2Center for Genetic Medicine Research, Children’s National Research Institute, Children’s National Hospital, Washington, DC 20012, USA; 3Institute of Medical Biochemistry and Molecular Biology, University Medicine of Greifswald, 17489 Greifswald, Germany; 4Friedrich-Baur-Institute, Department of Neurology, Klinikum der Universität, Ludwig-Maximilians-Universität, 80336 Munich, Germany; 5Department of Medicine IV, Klinikum der Universität, Ludwig-Maximilians-Universität, 80336 Munich, Germany

**Keywords:** optical genome mapping, OGM, chromosome analysis, CA, chromosomal microarray analysis, CMA, structural variant, SV, copy number variant CNV, chromosomal rearrangements

## Abstract

Chromosome analysis (CA) and chromosomal microarray analysis (CMA) have been successfully used to diagnose genetic disorders. However, many conditions remain undiagnosed due to limitations in resolution (CA) and detection of only unbalanced events (CMA). Optical genome mapping (OGM) has the potential to address these limitations by capturing both structural variants (SVs) resulting in copy number changes and balanced rearrangements with high resolution. In this study, we investigated OGM’s concordance using 87 SVs previously identified by CA, CMA, or Southern blot. Overall, OGM was 98% concordant with only three discordant cases: (1) uncalled translocation with one breakpoint in a centromere; (2) uncalled duplication with breakpoints in the pseudoautosomal region 1; and (3) uncalled mosaic triplication originating from a marker chromosome. OGM provided diagnosis for three previously unsolved cases: (1) disruption of the *SON* gene due to a balanced reciprocal translocation; (2) disruption of the *NBEA* gene due to an inverted insertion; (3) disruption of the *TSC2* gene due to a mosaic deletion. We show that OGM is a valid method for the detection of many types of SVs in a single assay and is highly concordant with legacy cytogenomic methods; however, it has limited SV detection capabilities in centromeric and pseudoautosomal regions.

## 1. Introduction

The completion of the human genome project marked the development of diagnostic technologies contributing toward uncovering the genetic causes of many disorders [1]. Compared to traditional cytogenetic technologies such as chromosome analysis (CA), novel high throughput genomic methods such as chromosomal microarray analysis (CMA) brought diagnostic scalability, increased resolution, and sensitivity in copy number variant (CNV) detection on a genomic scale.

While detection of CNVs by CMA has significantly improved the detection rate of chromosome anomalies, other types of structural variation (SV) such as balanced translocations or inversions cannot be detected by this technology, nor can CMA elucidate the structure of the CNVs it detects [2]. Identification of balanced events and the resultant genomic structure may provide crucial information needed for recurrence risk estimation. For example, a copy number gain, identified by CMA, may in fact be non-tandem and inserted elsewhere in the genome, potentially disrupting a gene. A balanced translocation identified by CA could disrupt a disease-causing gene not identifiable due to the nature and limitations of conventional cytogenetic approaches.

Optical genome mapping (OGM), a novel high-throughput cytogenomic method that identifies structural variants (SVs)—aneuploidies, insertions, deletions, duplications, inversions, translocations, repeat expansions/contractions, as well as copy neutral absence of heterozygosity [3,4,5,6]. OGM’s ability to detect this range of abnormalities is obtained via the utilization of long DNA molecules labeled at specific sequence motifs that can span most known repetitive elements and assembled into haplotype-resolved contigs for SV calling. OGM has been used for diagnosis in constitutional disorders [7,8,9,10,11] and cancers [12,13,14], showing that OGM is accurate in SV identification, and can have better turnaround time and cost efficiency than the combination of CA and CMA. Here, we validated OGM’s SV detection capabilities against 87 clinically identified SVs. We show that OGM is 98% concordant with variants identified by CA, CMA, or Southern blot. It provides an improved characterization of the genomic architecture of SVs. Its capability to investigate balanced chromosome alterations at high resolution leads to deciphering clinically relevant genetic causes of a patient’s phenotype.

## 2. Materials and Methods

### 2.1. Cohort Composition

A total of 76 patient samples with a diverse set of phenotypic indications referred for chromosome analysis, chromosomal microarray analysis, or repeat array unit and haplotype analysis in cases of FSHD1 were recruited for the study (Appendix A). The validation cohort was selected to include the following types of SVs: (1) duplications/insertions (greater than 30 kbp, including interstitial and terminal duplications); (2) deletions (greater than 30 kbp, including interstitial or terminal deletions, as well as mosaic whole chromosome loss); (3) translocations/inversions (balanced and unbalanced); and (4) Repeats (D4Z4 repeat array units in facioscapulohumeral muscular dystrophy 1, FSHD1). Refer to Table 1 for the number of tested SVs in each of the subcategories.

### 2.2. Chromosome Analysis and Chromosomal Microarray Analysis

Chromosome analysis was performed using standardized protocols. Chromosomal microarray analysis was performed after standard DNA extraction protocols using Infinium CytoSNP-850K Bead chip (Illumina, San Diego, CA, USA) according to the manufacturer’s protocol.

### 2.3. Southern Blotting for D4Z4 Repeat Length Analysis

DNA was digested with *Eco*RI, *Eco*RI + BlnI and *Apo*I (= Isoschizomer to *Xap*I) (*Eco*RI, *Apo*I: New England Biolabs, Frankfurt, Germany; *Bln*I: Takara Company Europe GmbH, Frankfurt, Germany) and separated by electrophoresis for 40 h at 1.2 V/cm on 0.5% agarose gels in 1 × TAE buffer. DNA was transferred to Hybond XL membranes (GE Healthcare, Freiburg, Germany) and hybridized with radioactively labeled probe p13E-11. Bands were then visualized by autoradiography. Fragment lengths were determined by comparison to a standard marker included in each run (l DNA-Mono Cut Mix; New England Biolabs, Frankfurt, Germany) [15].

### 2.4. Optical Genome Mapping

#### 2.4.1. DNA Extraction and Labeling

Ultra-high molecular weight (UHMW) DNA was extracted from frozen/fresh whole blood following manufacturer’s protocols (Bionano, San Diego, CA, USA). The white blood cells were first digested with Proteinase K, then DNA was precipitated with isopropanol, bound with a nanobind magnetic disk, and washed. UHMW DNA was resuspended in the elution buffer and quantified with Qubit Double Stranded DNA BR Assay kit (ThermoFisher Scientific, Waltham, MA, USA).

DNA labeling was performed following manufacturer’s protocols (Bionano, USA). Direct Labeling Enzyme 1 (DLE-1) reactions were carried out using 750 ng of purified UHMW DNA. The fluorescently labeled DNA molecules were imaged sequentially across nanochannels on a Saphyr instrument (Bionano, USA). Effective genome coverage >80× was achieved for all tested samples. Samples also met the following QC metrics: label density of ~15/100 kbp; filtered molecules N50 (≥150 kbp) ≥ 230 kbp; map rate ≥ 70% (except for several samples that had 65% map rate).

#### 2.4.2. Data Analysis

Automated, OGM-specific pipelines—Bionano Access and Solve (versions 1.7.1 and 3.7, respectively), were used for data processing. Specifically, two pipelines were used for variant identification: de novo assembly and fractional copy number analysis. De novo assembly was performed using Bionano’s custom assembler software program to generate consensus genome maps (*.cmap). SVs were identified based on the alignment profiles between the de novo assembled genome maps and the Human Genome Reference Consortium GRCh38. If the assembled maps did not align contiguously to the reference, then a putative SV was called. Fractional copy number analyses were performed from alignment of molecules and labels against GRCh38. A sample’s raw label coverage was normalized against relative coverage from normal human controls, segmented, and baseline copy number (CN) state estimated from calculating mode of coverage of all labels. Certain SV and CN calls were masked, if occurring in GRCh38 regions found to be in high variance (e.g., gaps, segmental duplications). Variants were filtered based on SV/CNV quality metrics, masking of difficult to align regions (e.g., centromeres, telomeres, reference gaps), SV call frequency, and CNV size.

### 2.5. Nanopore Sequencing

Next, 1.5 µg of genomic DNA was prepared following the SQK-LSK110 genomic DNA ligation protocol (Oxford Nanopore Technologies, Oxford, UK) and subsequently sequenced using an R9.4.1 (FLO-MIN106D) flow cell on a GridION Mk1 instrument, with a sequencing duration of 72 h. Adaptive sampling techniques were employed to target the specific gene and its associated flanking regions, collectively spanning approximately 30 Mbp or 1% of the genome in each case. After 30 h of runtime, the sequencing process was briefly paused for a flow cell wash (EXP-WSH004), followed by reloading with a second freshly prepared library, and then the sequencing continued for the remaining duration.

Genomic regions selected for nanopore adaptive sequence sampling included GRCh38 chr21:26041038-41079514; chr6:71870963-86870963 for *SON* gene translocation (case 61), and chr16:1-10988730 for *TSC2* mosaic deletion (case 11). Obtained long reads were mapped against the GRCh38/hg38 reference sequence using Minimap2 v2.17-r941. Structural variation detection was carried out using Sniffles v2.0.7 and NanoVar v1.3.8. The mapped data were further subjected to manual assessment using the Integrative Genomics Viewer 2.13.0 (IGV, Broad Institute, Cambridge, MA, USA).

## 3. Results

### 3.1. OGM Concordance in Detection of Chromosomal Abnormalities

OGM was attempted on a total of 76 patients containing 87 previously identified SVs. The surveyed phenotypes included developmental delay, intellectual disability, congenital anomalies, and facioscapulohumeral muscular dystrophy 1 (FSHD1). One patient presented with a very clear clinical presentation of Tuberous Sclerosis, yet undiagnosed with conventional testing (case 11, Appendix A). OGM provided the genetic diagnosis for this case (discussed later in the results). As a result, OGM concordance was calculated based on the residual number of 75 patients and 87 SVs, which were subdivided into four groups for individual calculations of concordance by variant type as well as overall concordance (Table 1).

**Table 1 genes-15-00342-t001:** Concordance between OGM and CA, CMA, Southern blot.

Method	Structural Variant Types				Total
	Duplication/Insertion	Deletion	Translocation/Inversion	Repeat (FSHD1)	
CA, CMA, SB	38	22	17	10	87
OGM	36 *	22 *	16	10	84
Concordance	95%	100%	94%	100%	98%

Abbreviations: CA—chromosome analysis, CMA—chromosomal microarray, SB—Southern blot. * A single duplication (case 39) and a single deletion (case 16) were classified as concordant upon manual review of the OGM data (Appendix A, respectively).

In total, 87 different SVs were investigated including 3.3 kbp D4Z4 repeat units in patients suspicious for facioscapulohumeral muscular dystrophy 1 (FSHD1), diagnosed clinically. The concordance for interstitial and terminal duplications/insertions, and deletions was 95% and 100%, respectively. Similarly, the concordance for translocations/inversions and D4Z4 repeat arrays was 94% and 100%, respectively. The overall OGM concordance with orthogonal laboratory methods was 98%.

### 3.2. Modality of Structural Variant Calling and Advantages of OGM

#### 3.2.1. Copy Number Variants

OGM successfully identified deletions and duplications previously detected by CA or CMA. The 15q11.2q13.1 recurrent microdeletion/microduplication SVs shown in Appendix A provide examples of how OGM can identify these types of variants (cases 9 and 40, respectively). Similar to CMA, where the copy number and B-allele frequency (BAF) tracks are used for cross validation of the called CNVs, OGM identifies deletions (copy number losses) and duplications (copy number gains) using two independent bioinformatics methods: (1) direct molecule alignments to the reference genome generating coverage pileups (CNV track), and (2) independent of the reference genome sample molecule assembly and comparison of the assembly to the reference (map track), (see methods). As Appendix A demonstrates, chromosome 15q11.2q13.1 deletion and duplication calls were identified by both the CNV and map tracks. Molecules spanning the fusion of the duplicated regions demonstrate that the duplication is in tandem. OGM predicted matchable sizes and breakpoints for both deletions and duplications within the limitations of each technology as published elsewhere [16]. The difference between size calls of the two methods CMA and OGM are typically due to the presence of low copy repeats (LCRs) flanking the recurrent microdeletions and microduplications.

#### 3.2.2. Balanced/Unbalanced SVs

One major advantage of OGM compared to CMA and CA is its capability to identify balanced rearrangements such as inversions or translocations with a precise localization of the breakpoints, thus enabling detection of affected genes. Appendix A demonstrates an example of an inversion identified by OGM with a much better resolution compared to CA allowing for the assessment of the involved genes (case 66).

Derivative chromosomes due to unbalanced translocations can be investigated in a single assay without the need of a follow-up targeted fluorescence in situ hybridization (FISH) analysis. As such, the example in Appendix A shows a derivative chromosome 4 that was natively identified by OGM with terminal deletion/duplication on chromosomes 4q35.1 and 12q24.33, respectively, as well as with the corresponding fusion between the CNV breakpoint locations indicating a translocation (case 54).

#### 3.2.3. Repeat Length Investigations

Measuring of repeat length in repeat contraction disorders such as facioscapulohumeral muscular dystrophy 1 (FSHD1) can precisely determine the number of D4Z4 repeats at the telomeric end of chromosome 4q35. OGM also discriminates between permissive and non-permissive haplotypes (Appendix A, case 69).

### 3.3. Genomic Structure Solved by OGM

#### 3.3.1. Size Determination of Balanced Events

Increased resolution of OGM allows for better assessment of sizes of genomic rearrangements. For example, Figure 1A, case 64 demonstrates the result obtained by OGM in a male with recurrent pregnancy losses of his partner and referred for preimplantation genetic testing. Previous testing by CA had identified a balanced reciprocal translocation of chromosomes 14 and 22, but neither the size of the exchanged material nor the possibility of a potential CNV at the breakpoint could be resolved. Using OGM, we were able to identify (1) the base pair locations of the translocation breakpoints to provide the accurate sizing of affected DNA, and that there were (2) no associated CNVs in the breakpoints.

#### 3.3.2. Resolving the Genomic Structure of Two Adjacent Copy Number Gains

In addition to providing better sizing and more accurate breakpoint localization for the identification of impacted genes, OGM is highly informative in providing the insertion locations of duplications. For example, case 21 was clinically diagnosed with developmental delay and cerebral aneurysm. CMA had identified two copy number gains in the long arm of chromosome 2, which were also detected by OGM. However, unlike CMA, the de novo assembled consensus genome maps and their corresponding alignments to GRCh38 allowed for the interpretation of the precise genomic position where copy number gains were inserted, thus allowing reconstruction of the underlying genomic structure (Figure 1B).

### 3.4. Novel Findings Provided by OGM

#### 3.4.1. Detection of a Reciprocal Translocation Breakpoint in a Clinically Relevant Gene

Detection of balanced genomic rearrangements is an important aspect of clinical diagnostics. Unlike chromosome analysis, OGM provides sufficient resolution needed to identify genes that may be impacted at the breakpoints of a balanced translocation, inversion, or insertion. We investigated 17 cases of balanced translocations and inversions with OGM, all but one (case 65) were concordant with karyotype findings. Moreover, in case 61 (Figure 2A) clinically diagnosed with severe global developmental delay, intellectual disability, and microcephaly, the precise breakpoints of a reciprocal translocation of chromosomes 6 and 21 were determined by OGM. It was shown that the breakpoint in chromosome 21q22.11 disrupts the *SON* gene (OMIM 182465). *SON* encodes a DNA binding protein in which de novo truncating variants have been shown to cause intellectual disability and congenital malformations, the likely diagnosis for this case [17,18]. Previous clinical testing of this sample included negative CMA, panel, and exome sequencing. In addition to the translocation, OGM identified a 28 kbp insertion at the breakpoint of chromosome 6q14.1 of unknown origin (Appendix A). Due to the calculated frequency of the insertion of ~80% in the Bionano Access healthy population, we hypothesized that it is a haplotype variability absent from the main chromosome assembly of GRCh38/hg38.

To validate the gene overlap and breakpoints, targeted nanopore-based long-read sequencing was performed, confirming the translocation and breakpoints of chromosomes 6 and 21 (Figure 2A). Specifically, the haplotype chr6_GL383533v1_alt was identified, resolving the content of the 28 kbp insertion shown by OGM. According to GENCODE v41, this region does not contain any known coding genes. On chromosome 21q22.11, the breakpoint was found within intron 6 of the *SON* gene (NM_138927.4), leading to gene disruption. With a sequencing depth of 32 uniquely mapping reads at the breakpoint in *SON* and 25 reads at the breakpoint on chr6_GL383533v1_alt, 19 reads were spanning the translocation, reflecting a fraction of 59% and 76%, respectively. The higher fraction is likely due to the haplotype sequence on the patch not particularly being enriched during adaptive sampling.

#### 3.4.2. Solving Structure of a Chromosome Aberration and Detection of a Clinically Relevant Gene in Breakpoint of an Insertion

The patient, case 51, presented in clinic with developmental delay, intellectual disability, and behavioral anomalies. Chromosome analysis detected an insertion of a segment of chromosome 13 into chromosome 18 without determining the exact breakpoints. CMA revealed a 1.39 Mbp copy number loss on chromosome 13q22.1q31.1, which did not explain the patient’s phenotype. Investigation of this case by OGM identified both the insertion and the copy number loss 13q22.1q31.1. Moreover, OGM revealed an inverted orientation of the inserted segment, and detected the second breakpoint of the inserted segment of chromosome 13 in band 13q13.3, which disrupts gene *NBEA* (OMIM 604889). Thus, OGM solved the resultant structure of both chromosomes and identified *NBEA* disruption as a potential cause of the patient’s phenotype (Figure 2B).

#### 3.4.3. Solving a Low Mosaic Loss of a Clinically Relevant Gene

The capability of identification of mosaic SVs is a key advantage of OGM and is crucial for providing genetic diagnosis. To prove the detection of mosaicism, we chose a case with a mosaic loss of the X chromosome, previously detected by CA and CMA, which was confirmed by OGM (Appendix A, case 20). More exciting, however, was a case (case 11) clinically diagnosed with Tuberous Sclerosis, and for which previous extensive testing of the *TSC1* and *TSC2* genes was unremarkable. OGM led to the identification of a mosaic deletion of *TSC2* and *PKD1* genes with an allele fraction of approximately 12% (Figure 2C).

The mosaic *TSC2-PKD1* contiguous gene deletion was confirmed using targeted nanopore-based long-read sequencing. Long-read sequencing revealed a *TSC2-PKD1* contiguous gene deletion of 32.6 kbp encompassing exons 31–42 of *TCS2* and exons 12–46 of *PKD1* (Figure 1A). With a sequencing depth of 32 uniquely mapping reads at the breakpoint in *TSC2* and 29 in *PKD1*, 6 reads were spanning the deletion, reflecting a fraction of 19% and 21%, respectively. Investigation of short-read exome sequencing showed a slight decrease in exon coverage of the *TSC2* gene; however, it did not pass the CNV cutoff threshold and was not called.

### 3.5. Discordant and Ambiguous Cases

#### 3.5.1. Discordant Cases

OGM was discordant for a reciprocal translocation identified by (1) chromosomal analysis, (2) a duplication identified by CMA, and (3) a mosaic triplication (marker chromosome 22) identified by CA and CMA (Appendix A, cases 65, 3, 33).

(1) OGM does not detect aberrations when either a single or both breakpoints occur in centromeric or pericentromeric regions, e.g., in the p-arm of an acrocentric chromosome. Case 65 demonstrates that a reciprocal translocation with breakpoints in chromosome 14p11.2 and 17q25.1 fails to be detected by OGM as Appendix A shows. The translocation was not evident from either the circos plot or manual investigation of assembled maps of the involved chromosomes.

(2) In case 3, a duplication detected by CMA was localized within pseudoautosomal region 1 (PAR1) where the region similarity leads to cross-map alignments between chromosomes X and Y resulting in reduced sensitivity, thus a missed call both in SV and CNV tracks.

(3) In case 33, a mosaic gain of two copies 22p11.2q11.22 of a marker chromosome 22 was not identified. The case also had two additional marker chromosomes derived from chromosomes 8 and 19, both of which were called by OGM.

#### 3.5.2. Ambiguous Cases

In case 42, OGM identified the origin of the marker chromosome 19, although, the mosaic level at which the marker was present could not be determined (Appendix A). Cases 3 and 33 indicate that while OGM may resolve the origin of marker chromosomes and provide hints that the identified copy number gain is a result of a marker, these findings need to be validated by orthogonal methods.

In duplication/deletion validations, OGM did not automatically call two CNVs identified by CMA, as Appendix A show (cases 39 and 16, respectively). In case 39, the CNV profile indicates a potential gain 15q11.2; however, the region is noisy, and the map assemblies show multiple alignments due to segmental duplications. SV identification in these regions poses difficulties. In case 16, both the CNV profile and map assembly indicate a deletion in pseudoautosomal region 1 (PAR1); however, this CNV loss was not called due to masking of the region. In general, the CNV profile of PAR1 region is noisy and distinguishing between a true variant versus noise could be challenging. However, both variants were visible after manual inspection of the OGM-assembled maps and CNV data plots.

## 4. Discussion

In this study, 87 SVs were analyzed to validate the utilization of OGM in a clinical setting (Appendix A). As demonstrated, OGM allows for the identification of most classes of genomic variations, showing 98% concordance with other cytogenomic methods and methods for the investigation of D4Z4 repeat unit contraction in FSHD1 patients (Table 1). Analysis of the OGM data not only successfully identified previously known variants, but also revealed the insertion locations and genomic structure for most of the duplications (Figure 1). Lastly, using OGM, we identified definitive diagnoses for three cases listed in Figure 2.

The underlying methods for SV size identification between CMA and OGM are different, which directly translates into a predictable discrepancy in the size of calls made between the two technologies. Hybridization of oligonucleotide probes in CMA, targeted throughout the human genome, results in regions with good probe coverage as well as regions where the probe coverage is scarce. In turn, this leads to CNV sizing limitations, particularly in repetitive regions. Sizing is dependent on the proximity of unaffected probes to the CNV breakpoints (i.e., CMA reports the minimum size for breakpoint locations of identified CNVs). In contrast, OGM relies on utilization of many long DNA molecules for genome assembly. This results in the ability to accurately measure the length of DNA at any given region of the genome (i.e., assembled map information provides accurate SV sizing within approximately 60 bp); however, the breakpoint locations are dependent on label density and show the largest possible coordinates (+/− 3.3 kbp) [19].

OGM has its limitations in the detection of SVs in centromeric regions or in repetitive sequence regions, i.e., short arms of acrocentric chromosomes and pseudoautosomal regions. Therefore, OGM cannot detect SVs if only one of the breakpoints is in a repetitive region of a chromosome. In contrast to conventional chromosome analysis, OGM is not able to detect Robertsonian translocations and centric fusions, a limitation which also applies to CMA. Thus, Robertsonian translocations and centric fusions should be investigated using chromosome analysis or fluorescence in situ hybridization (FISH, given the availability of probes).

To date, the major application for OGM are investigations of hematological malignancies where it proved to be the superior approach compared to conventional CA and FISH. Applications in constitutional cases have successfully been implemented in the diagnostic approach and are starting to display incremental benefits. OGM may be implemented alongside chromosomal analysis or in conjunction with sequencing methods, but its placement ultimately depends on the laboratory’s existing diagnostic menu. A drawback still is the limited number of curated SVs and the absence of reliable databases that can be used for variant interpretation, making pathogenicity classification time-consuming and demanding for a high-throughput laboratory.

## 5. Conclusions

As demonstrated, OGM has both the technical and analytical validity for providing genetic diagnoses in a clinical diagnostic procedure. OGM achieved 98% concordance with SVs identified by CA, CMA, or D4Z4 repeat unit and haplotype investigations. In a single assay, OGM identifies both balanced and unbalanced SVs, thereby mitigating the need of additional reflex testing for refinement or validation. Beyond the ability of OGM to make concordant SV calls, OGM can provide critical additional structural information that may be useful for variant curation. As such, OGM found insertion locations for most of the surveyed duplications, and it was instrumental in the identification of diagnoses in three cases. Some SVs identified by OGM may require advanced understanding of the method to solve the underlying genomic structure and/or additional investigation to delineate clinical significance. Growing database sets will help to contextualize OGM results.

## Figures and Tables

**Figure 1 genes-15-00342-f001:**
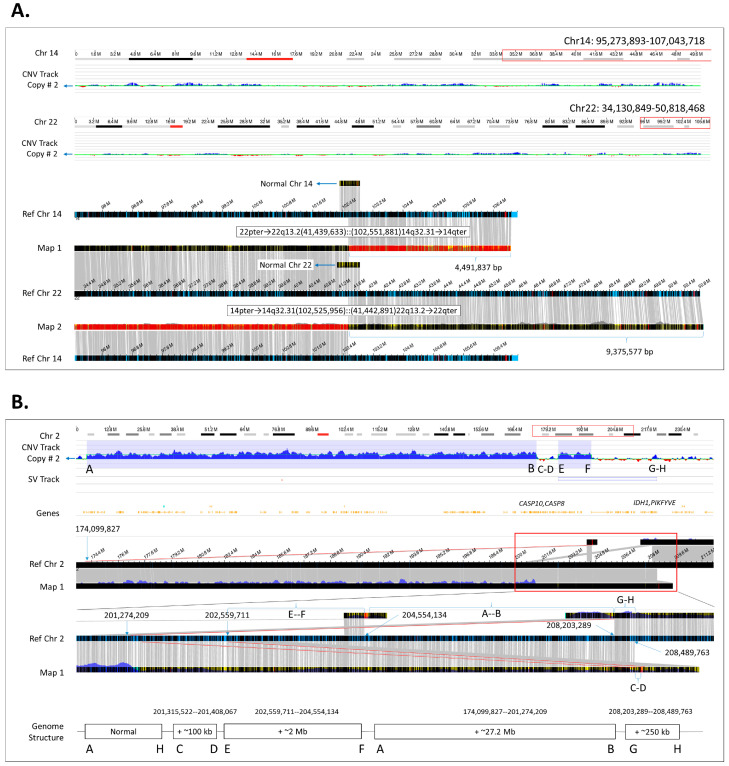
Genomic structure solved by OGM. (**A**) Defining the size of chromosomal segments in a reciprocal translocation carrier: case 64. Visualization of the balanced translocation by OGM copy number profiles for chromosomes 14 and 22 (top) and molecule map assemblies (bottom). As evident from the copy number variant track, there are no gains or losses throughout either chromosome. The map assemblies (map 1 and map 2) show alignments to both chromosome 14 and 22, indicating a reciprocal exchange of the terminal material between the q arms of chr14:102,525,956 and chr22:41,439,633 (4.5 Mbp and 9.4 Mbp in size, respectively). The resultant derivative chromosome compositions are written on top of maps 1 and 2. The smaller maps that span the breakpoints indicate the presence of a normal chromosome. (**B**) Deciphering the underlying genomic structure of two copy number gains: case 21. The OGM CNV track (top) shows two gains of chromosome segments 2q31.1q33.1 and 2q33.2q33.3, 27 Mbp and 2 Mbp in size, respectively. The zoom-in around the red rectangle shows the maps and the corresponding breakpoints that were used to solve the resultant genomic structure (bottom). Both duplications are inserted downstream in the same orientation.

**Figure 2 genes-15-00342-f002:**
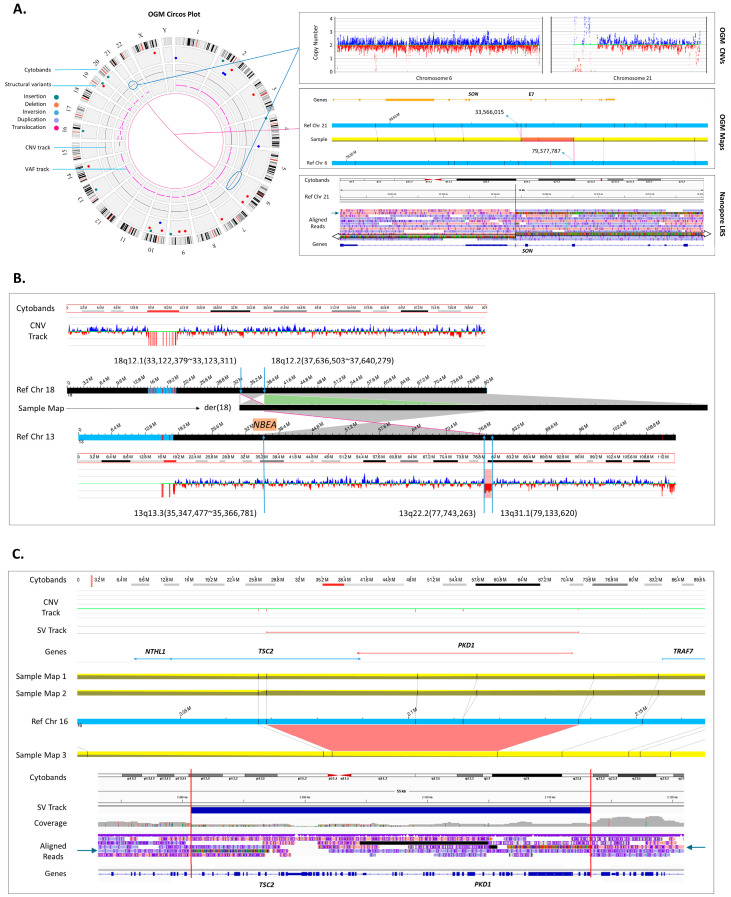
Novel findings provided by OGM. (**A**) Identification of disease-causing gene disruption due to a reciprocal translocation: case 61. OGM circos plot showing a fusion line between chromosome 6 and 21 (left). The corresponding CNV plots (top right) show no change in the copy number of either chromosome indicating that the exchange is likely balanced. The fusion is evidenced by an assembled map alignment to both chromosomes (middle right). The targeted long-read nanopore sequencing confirms the translocation by showing read alignments (arrows) on each side of the chromosome 21 breakpoint and misalignment in the middle (red line). Mismatched sequences from reads spanning the breakpoint map to 6q14.1 and the breakpoint on chromosome 21 disrupts *SON* gene (chromosome 21q22.11) within intron 6. (**B**) Identification of disease-causing gene disruption due to an inverted insertion: case 51. OGM CNV plots of both chromosomes 18 and 13 are demonstrated at the top and bottom. Chromosome 18 does not have any copy number changes; however, CNV plot of chromosome 13 shows CNV loss (chr13q22.2q31.1) adjacent to which there is an insertion breakpoint with inverted map alignment to chromosome 18. Another breakpoint on chromosome 13 (chr13q13.3) overlaps with *NBEA* gene which has a phenotypic overlap with the clinical presentation of the patient. Moreover, a segment on chromosome 18 that overlaps the insertion breakpoint is inverted but does not seem to overlap a clinically significant gene. The derivative chromosome 18 map assembled with OGM is shown in the middle with corresponding alignments to chromosomes 13 and 18. (**C**) Detection of a disease-causing gene disruption due to a mosaic deletion: case 11. OGM CNV plot shows no change in copy number around *TSC2* and *PKD1* genes; however, the OGM map assemblies show two normal alignments to the reference (maps 1 and 2) and, most importantly, a third map (map 3) that shows a deletion overlapping these two genes. The third map assembly indicates that the deletion is mosaic with an estimated molecule support of allele fraction at 12%. The OGM map indicates a wide range for breakpoint localization, requiring orthogonal validation to identify more precise deletion locations. Targeted nanopore sequencing showed that *TSC2-PKD1* contiguous gene deletion spans 32.6 kbp. Realignment of the mismatched sequences reveals the breakpoints within intron 30 of the *TSC2* gene and intron 11 of the *PDK1* gene.

## Data Availability

All data supporting this study are included with this paper with the exception of individual alignment and variant call files, and personally identifiable information to protect patients’ identities.

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
