# Peer review of "Optical Genome Mapping as a Potential Routine Clinical Diagnostic Method"

_genes, 2024, doi:10.3390/genes15030342_

Round 1

Reviewer 1 Report

Comments and Suggestions for Authors

The manuscript is clear, actual, and well documented. The authors demonstrate the advantages of optical genome mapping ( OGM ) as compared to chromosome analysis and CGH array methods .
Some minor remarks:

1. The cause of difference in number of patients (76) and gene variants (88) should be noted in the text. It is just clear after regarding Table S1, some(9) patients had several variations.

2. It would be interesting to see, how did change the genetic counselling after the more precise OGM results in the analyzed patients?

3. In the discussion some remark should be given on potential other method(s) which could detect SVs in centromeric and pseudocentromeric regions as well, where OGM is not helpful.
The high level manuscript is recommended for publication.

Author Response

Some minor remarks:

1. The cause of difference in number of patients (76) and gene variants (88) should be noted in the text. It is just clear after regarding Table S1, some(9) patients had several variations.

Thank you for the suggestion. We had also thought that this representation of the data may lead to small confusion in understanding the main concordance table. To solve it, we added the following sentence as the starting sentence of the results – “OGM was attempted on a total of 76 patients containing 87 previously identified SVs.” and discussion – “In this study, 87 SVs were analyzed to validate the utilization of OGM in a clinical setting”

2. It would be interesting to see, how did change the genetic counselling after the more precise OGM results in the analyzed patients?

We agree, unfortunately, the return of the updated results to the patients based on the newer report has not been tracked in this study.

3. In the discussion some remark should be given on potential other method(s) which could detect SVs in centromeric and pseudocentromeric regions as well, where OGM is not helpful.

We thank the reviewer for providing this suggestion. We added the following sentence in the discussion that addresses the comment. “Thus, Robertsonian translocations and centric fusions should be investigated using chromosome analysis or fluorescence in situ hybridization (FISH, given the availability of probes).”

The high level manuscript is recommended for publication.

We thank the author for the review, suggested comments and recommendation for publication of this study.

Reviewer 2 Report

Comments and Suggestions for Authors

Clinical genomic diagnostics are mainly divided into germline (mostly rare inherited disorders) and somatic groups (cancer). The content of this manuscript is based on identification of abnormalities in the germline, but this is not made clear in the introduction. Adding additional information to the introduction to emphasize that the current study evaluates OGM specifically for assessment of germline abnormalities would be useful.

Providing information on the sensitivity and limit of detection would be useful. Although the limit of detection is more relevant when this technique is applied to detect somatic abnormalities, this should still be addressed when investigating the sensitivity of OGM for detecting mosaicism.

Author Response

We agree. As the reviewer suggests, the limit of detection is an important aspect of OGM and is dependent on the obtained coverage of the reference genome. In this study, all samples had greater than 80X coverage which generally is sufficient to identify mosaicism in germline cases (depending on the type of assembly pipeline used) between 20-30%. However, we did not test OGM’s capabilities in LOD because it has been investigated and published previously by a different group – “Clinical Validation and Diagnostic Utility of Optical Genome Mapping for Enhanced Cytogenomic Analysis of Hematological Neoplasms” PMID: 36265723 DOI: 10.1016/j.jmoldx.2022.09.009 – OGM’s “limit of detection was determined to be at 5% allele fraction for aneuploidy, translocation, interstitial deletion, and duplication.”

Reviewer 3 Report

Comments and Suggestions for Authors

In the manuscript Barseghyan et al. the authors validated with Optical genome mapping (OGM) SV detection capabilities against 87 clinically identified SVs in a cohort of 76 patients. OGM reached a concordance as high as 98 % with variants identified by CA, CMA, or Southern blot. The investigated SVs encompassed duplications/insertions, deletions, translocations/inversions (balanced and unbalanced) and repeats. The results are presented clearly and in a structured way, illustrated by figures and supplementary figures with a detailed legend. Additionally, some particularly interesting cases, such as low mosaic loss of a clinically relevant gene and discordant and ambiguous cases are presented. One patient with a phenotype of Tuberous Sclerosis has remained undiagnosed after traditional testing and was diagnosed with OGM. The discussion presents the advantages and limitations of the method as well as current applications. Conclusions are concise and summarize the results.

Comments: 

1) It would be good to include 1-2 sentences about OGM methodology in the introduction, in addition to the description that is already provided in the discussion.

2) Are the statistics on the concordance from other studies available?

3) Please include in the discussion where you could place OGM in the current diagnostic schema.

Author Response

Comments: 

1) It would be good to include 1-2 sentences about OGM methodology in the introduction, in addition to the description that is already provided in the discussion.

We thank the reviewer for the suggestion and, initially, before the original submission we thought of providing a brief description of OGM in the introduction; however, later we decided against it because the method has been around since 2010 and the detailed description is available in the methods.

2) Are the statistics on the concordance from other studies available?

Yes, a much larger study across multiple clinical diagnostics laboratories in the United States compare OGM with other cytogenetics methods in 627 samples. The study had concordance rate of 98.6%, which is very similar to what we report. J Mol Diagn. 2024 Mar;26(3):213-226. doi: 10.1016/j.jmoldx.2023.12.003.

3) Please include in the discussion where you could place OGM in the current diagnostic schema.

We thank the reviewer for providing this suggestion. We added the following sentence in the discussion that addresses the comment. “OGM may be implemented alongside chromosomal analysis or in conjunction with sequencing methods, but its placement ultimately depends on the laboratory’s existing diagnostic menu.”
